# Predictive Value of Selected Plasma Biomarkers in the Assessment of the Occurrence and Severity of Coronary Artery Disease

**DOI:** 10.3390/ijms26020537

**Published:** 2025-01-10

**Authors:** Małgorzata Wojciechowska, Michał Nizio, Katarzyna Wróbel, Karol Momot, Katarzyna Czarzasta, Krzysztof Flis, Maciej Zarębiński

**Affiliations:** 1Chair and Department of Experimental and Clinical Physiology, Laboratory of Centre for Preclinical Research, Medical University of Warsaw, 02-097 Warsaw, Poland; malgorzata.wojciechowska2@wum.edu.pl (M.W.); michalnizio@outlook.com (M.N.); katarzyna.wrobel1608@gmail.com (K.W.); karol.momot@wum.edu.pl (K.M.); katarzyna.czarzasta@wum.edu.pl (K.C.); 2Department of Invasive Cardiology, Independent Public Specialist Western Hospital John Paul II, Lazarski University, 05-825 Grodzisk Mazowiecki, Poland; krzfl1977@gmail.com

**Keywords:** atherosclerosis, circulating biomarkers, interleukin-8

## Abstract

Despite significant advances in imaging modalities for diagnosing coronary artery disease (CAD), there remains a need for novel diagnostic approaches with high predictive values and fewer limitations. Circulating biomarkers, including cytokines such as interleukin-6 (IL-6) and interleukin-8 (IL-8), cell adhesion molecules such as soluble vascular cell adhesion molecule-1 (sVCAM-1), peptides secreted by endothelial cells such as endothelin-1 (ET-1), and enzymes involved in extracellular matrix remodeling such as a disintegrin and metalloproteinase with thrombospondin motifs-1 (ADAMTS-1) offer a promising alternative. This study aimed to evaluate the correlation between the plasma levels of selected biomarkers and the presence and severity of CAD. We enrolled 40 patients admitted for elective coronary angiography. CAD was defined as having at least one coronary artery stenosis ≥ 50%. The severity of CAD was assessed using the Gensini Score (GS). IL-8 levels were significantly higher in the CAD group, with a mean of 9.78 (SD 0.46) compared to 8.37 (SD 0.40) in the non-CAD group (*p* = 0.0228). No significant differences were observed for the other biomarkers between the groups. A positive Spearman correlation was found between IL-8 levels and the GS (ρ = 0.39, *p* = 0.017). These findings suggest that IL-8 may have potential as an additional tool for diagnosing or excluding atherosclerosis. Further studies with larger sample sizes are needed to validate its clinical utility.

## 1. Introduction

Atherosclerosis is a chronic inflammatory condition that affects the arterial walls and serves as the pathological foundation for numerous cardiovascular diseases. The progression of atherosclerosis is intricately linked to inflammation, which plays a central role in both the initiation and development of the disease [1]. When atherosclerosis impacts the coronary arteries, it leads to coronary artery disease (CAD), a significant cause of morbidity and mortality worldwide. Diagnosing CAD typically involves evaluating clinical symptoms and using imaging and functional studies, which help determine the presence and severity of coronary artery stenosis.

Coronarography is an invasive examination that carries the risk of complications, mainly related to the vascular access site [2]. Computed tomography of coronary arteries (CCTA) is a non-invasive examination, but the same as coronarography, involves exposure to ionizing radiation. In addition, both types of examinations involve contrast media, which may be harmful in patients with kidney and thyroid diseases. Functional tests, such as Positron Emission Tomography (PET), Single-Photon Emission Computerized Tomography (SPECT), stress echocardiography, or a treadmill test, are also used to diagnose CAD. The above tests are performed depending on the likelihood of CAD and the presence of symptoms, as well as their indications, contraindications, possible complications, and limitations [3].

An interesting trend involves attempts to diagnose atherosclerosis based on laboratory tests. So-called ‘biomarkers of atherosclerosis’ have been sought for years. These are cytokines, cellular adhesion molecules, and other markers of inflammation and oxidative stress responsible for the initiation and progression of atherosclerotic plaque. In theory, measuring the concentration of these substances in the plasma could be used to diagnose atherosclerosis, assess its advancement, or determine cardiovascular risk more precisely [4]. Research has been going on for years, and some results are very optimistic, even regarding using these biomarkers as therapeutic targets [5,6,7,8,9]. However, there are also many ‘negative’ studies, so ultimately, there are no clear conclusions [9,10], and the determination of circulating biomarkers is still not included in the CAD diagnostic guidelines.

Pro-inflammatory cytokines participate in all disease steps, from early endothelial dysfunction to the late formation and disruption of vulnerable atherosclerotic plaque [11,12,13]. Several risk factors, such as genetic predisposition, high blood levels of low-density lipoproteins (LDLs), accumulation of LDLs in the arterial wall, followed by their oxidation (oxLDLs), high blood pressure, hyperglycemia, and smoking, play crucial roles in initiating and enhancing the disease process. These factors activate endothelial cells, promote the expression of adhesion molecules, and facilitate the migration of monocytes into the arterial wall. Once inside the arterial wall, monocytes differentiate into macrophages, accumulating oxLDLs. This accumulation triggers the production of various cytokines by macrophages. Among the cytokines involved in the process of atherogenesis, an important role is assigned to interleukin-6 (IL-6) and interleukin-8 (IL-8) [8,14,15].

Soluble vascular cell adhesion molecule-1 (VCAM-1) is an inducible glycoprotein and predominantly expressed in endothelial cells. Its expression is activated by pro-inflammatory cytokines, endothelin-1 (ET-1), reactive oxygen species, oxLDLs, hyperglycemia, and shear stress. Once expressed on the endothelial cell surface, VCAM-1 can bind to several ligands on leukocytes. This binding leads to cytoskeletal remodeling, disruption of endothelial cell junctions, and facilitation of transendothelial migration of leukocytes. Infiltration of monocytes into the arterial wall initiates the formation and development of atherosclerotic plaque [16].

ET-1 is a peptide secreted primarily by vascular endothelial cells. It has a potent vasoconstrictor and mitogenic properties and has been implicated in endothelial dysfunction, inflammation, and vascular remodeling. These include stimulation of cellular proliferation, synthesis of matrix proteins, and chemotactic effects on monocytes. ET-1 has increased the release of IL-6, IL-8, and other inflammatory cytokines from macrophages [17,18]. Thus, ET-1 may amplify and sustain macrophage activation in the developing atheromatous plaques. Ox-LDL results in increased ET-1 expression in cultured endothelial cells, and high ET-1 levels could promote further ox-LDL uptake in human endothelial cells [19]. Nonselective ET-1 receptor antagonists substantially inhibited the development of atherosclerosis in a genetic model of hyperlipidemia, possibly by inhibiting macrophage foam cell formation [20].

Recently, studies have shown the significant role of a disintegrin-like and metalloproteinase with thrombospondin motif (ADAMTS) in atherosclerosis progression. ADAMTS is a family of extracellular enzymes that regulate the structure and functioning of the extracellular matrix (ECM). The main components of ECM are proteoglycans (PG), the excessive degradation of which creates a space for pathological migration and proliferation of smooth muscle cells of blood vessels within the artery wall. Moreover, pathological degradation of ECM stimulates vascular wall infiltration by immune cells, intensifying the inflammatory process within the atherosclerotic plaque. According to the literature, most ADAMTS (including the most proteolytically active ADAMTS-1 and ADAMTS-4) are responsible for PG degradation and, thus, are atherosclerosis development [21,22,23,24,25,26,27,28,29,30,31]. Macrophages, foam cells, and smooth muscle cells are major contributors. In the mouse model, it has been shown that ADAMTS-1 cleaves ECM such as versican, which intensifies the inflammatory process within the atherosclerotic plaque, reducing its stability and increasing the risk of its rupture [21]. Human studies also confirm the involvement of ADAMTS-1 in developing unstable atherosclerotic plaque. For example, in the Pelisek J et al. study, there was significantly higher expression of ADAMTS1 in unstable carotid lesions than in stable plaques [24].

This study aimed to investigate a possible relationship between the circulating biomarkers sVCAM-1, ET-1, ADAMTS-1, IL-6, and IL-8 and the presence and extent of coronary lesions in stable patients admitted to the hospital for planned coronary angiography. In the context of ADAMT-1, this was a pilot study. The results may contribute to developing a non-invasive model for atherosclerosis diagnosis and help assess cardiovascular risk more accurately.

## 2. Results

### 2.1. Baseline Characteristics

The study included 40 patients admitted for scheduled coronary angiography due to stable angina, who were divided into two groups: a CAD group (n = 20), with at least one coronary artery stenosis ≥ 50%, and a non-CAD group (n = 20), including patients with normal coronary arteries or with mild intimal thickening. The two groups did not differ in terms of age, gender, and the presence of atherosclerosis risk factors, except for smoking, which was more common in the non-CAD group. In the non-CAD group, 16 patients had normal coronary arteries—0 points in Gensini Score (GS)—and 4 patients had mild intimal thickening, which could give the patients some points depending on the location and according to GS. Table 1 summarizes the baseline characteristics of the patients in both groups.

### 2.2. Circulatory Biomarkers

The study assessed the levels of various plasma biomarkers and their correlation with the presence and severity of CAD. The results of this study suggest that among the biomarkers examined, only IL-8 stands out as a significant indicator of CAD presence (non-CAD M 8.37, SD 0.40 vs. CAD M 9.78, SD 0.46, *p* = 0.0264). Other biomarkers, such as ET-1, sVCAM-1, ADAMTS-1, and IL-6, did not significantly differ between the CAD and non-CAD groups (Figure 1). A positive correlation between plasma levels of IL-8 and the advancement of CAD assessed by the GS was found (Spearman’s rho = 0.39, 95% CI: 0.07 to 0.63, *p* = 0.017; Figure 2).

Post hoc analysis revealed no significant differences in biomarker levels based on obesity, hypertension, chronic kidney disease, atrial fibrillation, or gender. However, one exception was observed for sVCAM-1, which showed significantly lower levels in patients taking statins (Mean: 5.66, SD: 8.43) compared to those not taking statins (Mean: 17.59, SD: 19.86; *p* = 0.01).

## 3. Discussion

The aim of our study was to determine circulating markers involved in the process of plaque formation at various stages of atherosclerosis development. We chose the relatively little studied sVCAM-1 and ET-1. Moreover, we decided on ADAMTS-1, which to our knowledge, no one has studied so far in the context of atherosclerosis. After analyzing the literature, we selected two promising cytokines: IL-6 and IL-8.

The results of our study indicate that among all the examined biomarkers, IL-6, IL-8, sVCAM-1, ADAMTS-1, and ET-1, only circulating IL-8 levels were significantly correlated with the presence and severity of CAD. This finding aligns with the study by Romuk et al., which demonstrated that IL-8 levels were elevated in patients with stable CAD compared to healthy controls and suggested that IL-8 could serve as a useful clinical predictor of unstable CAD [7]. In the study of Zhujun et al., patients with atherosclerosis exhibited higher serum levels of IL-8 than healthy controls; however, the study concerned patients with peripheral atherosclerosis, assessed by brachial–ankle pulse wave velocity, and one of the exclusive criteria was diabetes [32]. IL-8 has also been evaluated in the context of restenosis following stent implantation. In a study by Xia Li, it was found that IL-6, IL-8, hypercholesterolemia, diabetes mellitus, and high-sensitivity C-reactive protein (hs-CRP) were independent predictors of restenosis risk in CAD patients who underwent percutaneous coronary intervention with drug-eluting stents [33]. These findings suggest IL-8 could be crucial in predicting adverse outcomes in CAD patients post-intervention. However, there are also studies with contrasting findings. In an assessment of serum levels of several interleukins (IL), like IL-1β, IL-2, IL-4, IL-5, IL-6, IL-8, IL-10, IL-12p70, IL-17, tumor necrosis factor-α (TNF-α), interferon-α (IFN-α), and interferon-γ (IFN-γ) in patients suspected of having CAD who underwent coronary angiography, IL-8 levels did not differ significantly between those with no, mild, or severe CAD. Among all the cytokines examined, only IL-12p70 and IL-17 were identified as independent predictors of severe CAD [10]. Similarly, in the study of Kraaijenhof et al., no association was found between IL-8 plasma levels and plaque progression in CCTA over 12 months in stable CAD patients [34]. In the study by Inoue et al., none of the examined cytokines (IL-1, IL-2, IL-4, IL-5, IL-6, IL-8, IL-10, and TNF-α) showed a direct correlation with the severity of atherosclerosis assessed by GS. However, IL-8 was the only cytokine independently predicting long-term cardiovascular outcomes [9].

Up to this point, ADAMTS-1 levels have been shown to correlate with the presence of aortic aneurysms [35]. To our knowledge, our study is the first to investigate the levels of ADAMTS-1 in the blood, explicitly concerning CAD. While the role of ADAMTS-1 in the pathogenesis of atherosclerotic plaques has been well established, our results revealed no correlation between the presence or severity of CAD and the circulating ADAMTS-1 levels. The lack of differences could be explained by the small size of the study groups but also the relatively low advancement of atherosclerosis in the CAD group, as our patients had a low GS when compared to other studies (21 points in our study vs. 60 points in the previously mentioned study of Liu et al. [10] or 35 points in the study of Zhou M [36]). ADAMTS-1 release may be associated with a larger volume of atherosclerotic plaques. Moreover, it may have a more significant role in the pathogenesis of lesions with increased inflammation, active remodeling, and a tendency to rupture [24], which was not the subject of our study and requires further research. Similarly to ADAMTS-1, in the cases of ET-1, IL-6, and sVCAM-1, we did not demonstrate any correlation with the presence or severity of CAD. The current literature on their utility in diagnosing CAD is inconsistent. There are studies that both support and oppose their roles as diagnostic tools. Conflicting results of the presented report with studies showing a positive correlation of ET-1, sVCAM-1, and IL-6 concentration with atherosclerotic burden might find a similar explanation as in the case of ADAMTS-1, namely the fact that the analyzed CAD group was relatively healthy with low GS. The lack of correlation between the mentioned markers and the presence and severity of CAD may also result from the fact that the examined markers are not specific to atherosclerosis and are involved in many physiological processes. For example, sVCAM-1 and ET-1 are biomarkers for endothelial dysfunction in the course of arterial hypertension, obesity, and diabetes, whose prevalence was similar in both groups [37,38,39,40,41,42,43]. Also, the higher prevalence of smoking in the non-CAD group could have influenced the results (smoking and ET-1 [44,45]). Additionally, the non-CAD group included patients with intimal thickening of the coronary arteries with probable appearance of sVCAM-1 on the surface of the endothelium and the synthesis of ET-1 by dysfunctional endothelial cells. Research show that serum levels of sVCAM-1 might allow for the detection of endothelial activation at early stages of asymptomatic atherosclerosis [46,47]. Similarly, Lerman et al. demonstrated that ET-1 is a participant and marker for coronary endothelial dysfunction in the early stage of the disease [48]. Hence, the presence of the initial phase of atherosclerosis in some of the patients in the non-CAD group and the relatively low advancement of atherosclerosis in the CAD group could have resulted in similar results of the concentrations of the above markers in both groups. Some researchers suggest that IL-6, a pro-inflammatory cytokine, may be a marker for lesion instability and better estimate the risk of acute coronary syndrome [10,49], which we did not evaluate.

The table below summarizes the most significant studies from the last five years (2019–2024), both supporting and challenging the relevance of ET-1, IL-6, IL-8, and sVCAM-1 in CAD diagnosis and cardiovascular risk (Table 2).

To summarize the results of our study, we found a statistically significant association between the circulating IL-8 levels and the presence of CAD. This result was obtained despite the relatively low GSs in our patients. Additionally, a correlation was observed between IL-8 concentration and the advancement of CAD. It suggests that this cytokine is very sensitive marker of atherosclerosis, and its production increases from the early stages, and proportionally, with the progression of the disease. However, after a thorough review of the literature on this topic, it appears that the clinical utility of a single marker must be further explored. Statistical significance for the rest of the biomarkers, namely ET-1, aVCAM-1 ADAMTS-1 and IL-6, was not achieved in our study, which might be due to several factors, like the stable nature of CAD in the studied population, similar prevalence of risk factors like diabetes, obesity and hypertension in both groups, higher prevalence of smoking in the non-CAD group, the presence of intimal thickening of the coronary arteries in some patients from non-CAD group, and finally a relatively low advancement of atherosclerosis in CAD group. However, as there is a lot of encouraging data coming from other studies with larger study groups, we believe that the role of the mentioned particles in assessing the severity of atherosclerosis and estimating cardiovascular risk should also be further investigated. We deem that future of clinical practice may involve creating a probability calculator that integrates selected serum marker levels and the dynamics of the concentration, together with atherosclerotic risk factors and patient symptoms to enhance diagnostic accuracy and risk assessment. Another plausible approach is that serum biomarkers might be better suited for excluding rather than diagnosing atherosclerosis. In this scenario, a negative IL-8 results in a patient without other risk factors and a normal ECG could point to a non-atherosclerotic cause of chest pain and may be used as a non-invasive method to exclude healthy individuals before undergoing coronary angiography. This hypothesis, however, also warrants additional investigation.

### Limitations of the Study

The primary limitation of this study was the small sample size, which restricted our ability to combine biochemical results with atherosclerotic risk factors and perform multivariable analysis. Moreover, the study was a part of larger project, and the blood was collected two hours after coronary angiography [73], so it cannot be ruled out that it could have influenced the level of the tested substances. Additionally, the relatively low GS in CAD group might seem like a limitation, but this reflects the specific nature of this scoring system. A patient with a single critical stenosis may have a lower GS compared to a patient with multiple milder stenoses, even though the former may carry a higher cardiovascular risk. This disparity highlights the limitations of using GS alone to fully capture a patient’s risk profile.

## 4. Materials and Methods

The study received positive approval from the Bioethics Committee at the Medical University of Warsaw (KB/167/2020). It was conducted in the Department of Invasive Cardiology. Patients aged ≥18 years old scheduled for elective coronary angiography were invited to participate in the study if they met the inclusion criteria, namely the absence of chronic kidney disease (eGFR < 30 mL/min/1.73 m^2^), malignancy, or inflammatory diseases such as infections and autoimmune disorders. The patient was assigned to the appropriate group after the coronary angiography was performed and described by the interventional cardiologist.

Forty patients who met the inclusion criteria and provided written informed consent were included. Coronary angiography was performed using either a radial or distal radial approach. The amount of injected contrast and the total radial dose were recorded for each procedure. The patients were stable, and none of them required simultaneous stent implantation.

All procedures were performed by European Association of Percutaneous Cardiovascular Interventions (EAPCI)-certified operators without using ultrasound guidance.

All coronary arteries with a diameter sufficient for reliable visual analysis were assessed, and patients were categorized into two groups: the non-CAD and CAD. The non-CAD group consisted of patients with normal coronary arteries or with mild intimal thickening of the coronary walls (n = 20), and the CAD group included patients with at least one coronary artery stenosis ≥ 50% (n = 20), as 50% stenosis is usually the cut-off point for hemodynamically significant lesions [74]. GS was calculated for all patients to more accurately characterize the burden and severity of CAD [75].

Two hours after the coronary angiography, just after removing the pressure dressing, 10 mL of blood was collected from the cephalic vein into tubes with EDTA-K2 anticoagulant. The blood was centrifuged at 1600× *g* for 15 min at 4 °C. Then, the plasma was separated, immediately frozen, and stored at −20 °C until biochemical assessment was carried out at the Department’s Laboratory and the Department of Experimental and Clinical Physiology of the Center for Preclinical Research, Medical University of Warsaw.

Plasma concentrations of the selected markers were analyzed using the ELISA method with the following kits: Endothelin-1 ELISA Kit (Quantikine, DET100; R&D Systems, Inc., Minneapolis, MN, USA), Human IL-6 ELISA Kit (EH0201; FineTest Biotech Inc., Wuhan, China), Human ADAMTS1 ELISA Kit (ab213751; Abcam, Cambridge, UK), Human IL-8 ELISA Kit (KHC0081; Invitrogen, Waltham, MA, USA), and Human sVCAM-1/CD106 ELISA Kit (MBS2505831; MyBioSource, San Diego, CA, USA). To evaluate IL-6 concentration, plasma samples were diluted twice with a Sample Dilution Buffer. For the evaluation of the concentration of other plasma markers, they were not diluted. The ELISA tests were performed according to the manufacturer’s protocol. Each well’s optical density (OD value) was determined at once with an iMark™ Microplate Absorbance Reader (Bio-Rad Laboratories, Inc., Hercules, CA, USA) set to 450 nm. The concentration of each marker tested was estimated based on the standard curve. Microplate Manager^®^ 6 Software (Bio-Rad Laboratories, Inc.) was used to analyze the results.

The sample size was estimated using G*Power 3.1.9.7 software [76] and research outcomes from analysis of the available studies [7,43,51,64]. Power analysis indicated the following minimum sample size of chosen biomarkers assuming a value of 0.9 power and a significance level of 0.05: IL-8 minimum sample size < 30 to have an effect size d 1.224, IL-6 minimum sample size 30 to have an effect size d 1.135, sVCAM-1 minimum sample size 5 to have an effect size d 12.077, ET-1 minimum sample size < 20 to have an effect size d 1.570. Due to the planned ADAMTS-1 pilot study, it was decided that the total study group would be 40 subjects, i.e., 20 in each subgroup (CAD and non-CAD).

Statistical analysis was performed using Statistica software version 13.3. Categorical variables are expressed as numbers and percentages. Quantitative variables are presented as mean (M) with standard deviation (SD) for normally distributed variables or median (Me) with interquartile range (IQR) for non-normally distributed variables. Statistical significance was set at 0.05. The Shapiro–Wilk test was used to assess the normality of data distribution. The homogeneity of variances was evaluated using Levene’s test. For comparisons between the two groups (CAD vs. non-CAD), Student’s *t*-test was applied for normally distributed variables with equal variances, and the Mann–Whitney U test was used for non-normally distributed variables. Categorical variables were compared using the χ^2^ test. Correlations were analyzed between Gensini Score and IL-8 levels, and Spearman’s rho coefficient was estimated.

A pairwise deletion approach was used for missing data, meaning that only the available data for each specific analysis were included. Due to this study’s exploratory nature, correction for multiple comparisons was not applied. Therefore, the results should be interpreted cautiously, and the observed statistical significances require validation in further studies with larger sample sizes.

## 5. Conclusions

The main finding of the presented study is that IL-8 may have the potential to be a very sensitive tool for diagnosing or excluding atherosclerosis. However, further studies with larger sample sizes are needed to validate its clinical utility. Even though we did not achieve statistical significance for the remaining substances determined, there is a lot of encouraging data from other studies, so we believe the role of the mentioned particles in assessing the severity of atherosclerosis and estimating cardiovascular risk should also be further investigated. We believe future studies could focus primarily on evaluating the mutual correlations and concentrations’ dynamics between different biomarkers, as some are released into the bloodstream constantly and proportionally to atherosclerotic plaque size, and some are released in larger quantities only when atherosclerotic plaque becomes unstable. Such an approach could help predict acute coronary events more accurately by identifying patterns in biomarker interactions that precede the onset of plaque instability.

## Figures and Tables

**Figure 1 ijms-26-00537-f001:**
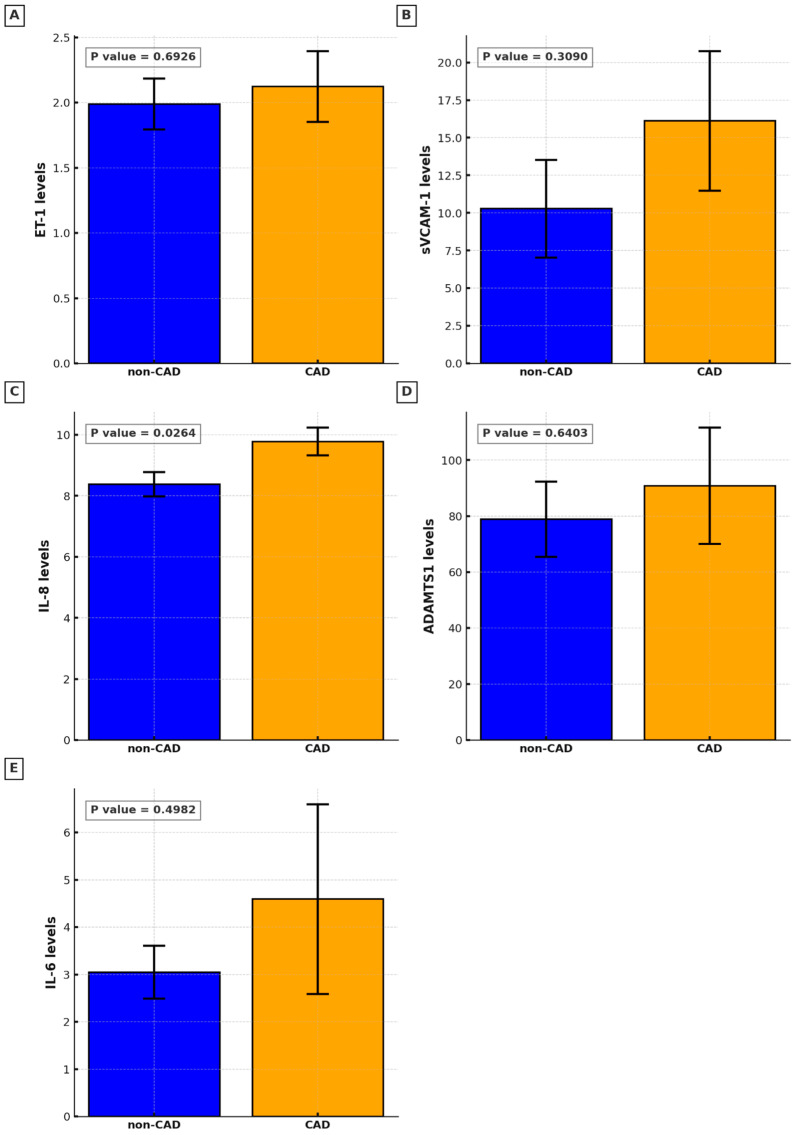
Circulating biomarkers levels in non-CAD and CAD groups: (**A**). Endothelin-1 (ET-1); (**B**). soluble vascular cell adhesion molecule-1 (sVCAM-1); (**C**). Interleukin-8 (IL-8); (**D**). a disintegrin-like and metalloproteinase with thrombospondin motif 1 (ADAMTS-1); (**E**). Interleukin-6 (IL-6).

**Figure 2 ijms-26-00537-f002:**
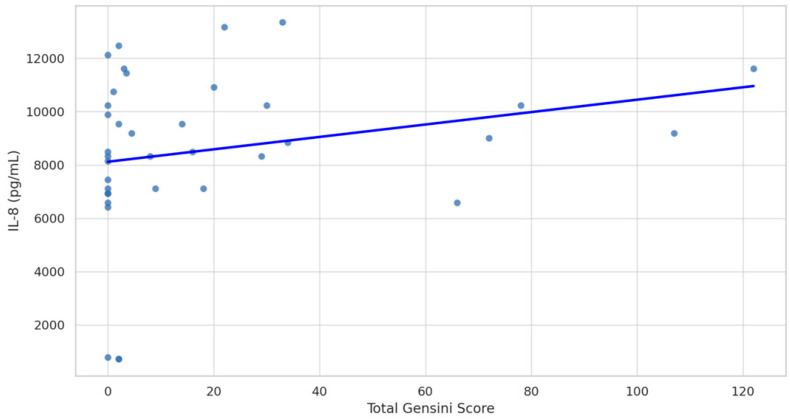
Spearman correlation between Gensini Score and Interleukin-8 (IL-8) levels. Spearman’s rho = 0.39 (95% CI: 0.07 to 0.63), *p*-value = 0.017.

**Table 1 ijms-26-00537-t001:** Baseline Characteristics.

Characteristic	CAD (n = 20)	Non-CAD (n = 20)	*p* Value
Male, n (%)	15 (75%)	12 (60%)	ns
Age, years (SD)	69 ± 1.75	67 ± 2.11	ns
Obesity, n (%)	7 (35%)	8 (40%)	ns
BMI, mean (SD)	28.72 ± 5.32	29.73 ± 5.76	ns
Diabetes mellitus, n (%)	5 (25%)	9 (45%)	ns
Kidney disease, n (%)	2 (10%)	0	ns
Hypertension, n (%)	11 (55)	13 (65%)	ns
Smoking, n (%)	5 (25%)	12 (60%)	0.03
Atrial fibrillation, n (%)	2 (10%)	5 (25%)	ns
Statin treatment, n (%)	5 (25%)	11 (55%)	ns
Total Gensini Score, median (IQR)	21 (7.5–42)	0 (0–0.25)	<0.0001

**Table 2 ijms-26-00537-t002:** Markers and their roles in predicting CAD (2019–2024).

Marker	Results
IL-8	IL-8 levels were higher in patients with stable CAD in comparison with healthy subjects. Additionally IL-8 appeared to be a useful clinical predictor of unstable CAD [7]
	None of the 10 cytokine levels, namely IL-1β, IL-2, IL-4, IL-5, IL-6, IL-8, IL-10, tumor necrosis factor (TNF)-α, granulocyte-macrophage colony stimulating factor (GM-CSF) and interferon-γ (IFN-γ), were correlated with the severity of CAD; however, IL-8 was the only cytokine that predicted long-term cardiovascular outcomes independently of the other cytokines and hs-CRP [9]
	IL-6, IL-8, hypercholesterolemia, diabetes mellitus, and hs-CRP independently predict restenosis risk in CAD patients who underwent percutaneous coronary intervention with drug-eluting stents [33]
	Patients with atherosclerosis exhibited higher serum levels of IL-8 than healthy controls [32]
	IL-8 serum level did not change significantly between severe and non-severe CAD patients. IL-12p70 and IL-17, HDL-C, gender and diabetes were the independent predictors of severe CAD [10]
	The lack of association between IL-8 plasma levels and plaque progression in CCTA over a 12-month period in stable CAD patients [34]
sVCAM-1	Premature CAD (PCAD) patients had significantly higher circulating values of sVCAM-1 that healthy controls; also, older CAD patients showed higher levels of sVCAM-1, CRP, and IL-2 when compared to their age-matched controls. However, after adjusting for multiple parameters, only CRP remained significant for PCAD and IL-2 remained significant for CAD [50]
	A significant elevation of serum levels of sVCAM-1 along with sICAM-1, E-selectin, ox-LDL and 8-iso-PGF2α in obese with atherosclerosis compared with obese without atherosclerosis or control groups [51]
	Compared with controls, metabolic syndrome (MetS) patients had higher prevalence of carotid plaques, which was associated with a remarkable increase in circulating sICAM-1, sVCAM-1 and PAI-1. The increase in sICAM-1, sVCAM-1 and PAI-1, together with decreases in omentin-1, pointed to the presence of subclinical atherosclerosis and improved CVD risk stratification in non-smoking patients at early stage MetS beyond the traditional scores [52]
	The lack of association between VCAM-1 plasma levels and plaque progression in CCTA over a 12-month period in stable CAD patients [34]
ET-1	An association of circulating ET-1 levels with higher risk for all-cause mortality, cardiovascular death, non-cardiovascular death and sudden cardiac death in patients with stable CAD (prognostic value) [53]
	Baseline high ET-1 levels were independently associated with long-term all-cause death in prediabetes and diabetes patients with CAD undergoing PCI, suggesting that ET-1 may be a valuable marker in patients with impaired glucose metabolism [41]
	ET-1 level was significantly higher in CAD patients than in controls. Increased ET-1 level was significantly associated with diabetes mellitus and dyslipidemia in patients with CAD [42]
	Increased plasma ET-1 was independently associated with a higher risk of adverse cardiovascular prognosis in patients with in-stent restenosis and diabetes (a predictive biomarker). However, high ET-1 was not significantly associated with the risk of major adverse cardiovascular events in patients without diabetes [54]
	ET-1 plasma level was significantly higher in the group of hypertensive patients with atherosclerosis in comparison with the other groups, especially hypertensive patients without atherosclerosis [43]
	ET-1 could be an independent predictor for the presence of noncalcified and mixed plaques, which are considered as the high-risk coronary plaques [55]
	Plasma ET-1 levels correlated with the severity and progression of CAD and associated with the need for revascularization in DM patients [56]
IL-6	Higher levels of IL-32, IL-36, TNF-α and IL-6 in the CAD group compared to control [57]
	Concentrations of hs-cTn and IL-6 were associated with CAD characteristics and MACEs.
	hs-cTn was associated with high-risk plaque and IL-6 with significant stenosis. In participants with nonobstructive CAD (stenosis 1–69%), the presence of both hs-cTn and IL-6 above median was strongly associated with MACEs [58]
	No association between IL-6 and the severity of CAD [10]
	There was a significant positive correlation between IL-6 and the severity of CAD assessed by Gensini score [59]
	IL-4, IL-6, and HDL-C levels were strongly associated with chronic total occlusion, and IL-6 was also linked to procedural outcomes of CTO [60]
	IL-6 levels positively correlated with the advancement of coronary artery disease and long-term survival [61]
	Circulating levels of IL-6 along with sICAM-1, sVCAM-1, E-selectin, ox-LDL and 8-iso-PGF2α are sensitive markers for early prediction of atherosclerosis in obese subjects [50]
	CAD patients, especially those with diabetes, have higher levels of IL-6 as compared with controls [62]
	After adjusting for age, sex, LVEF, ischemia severity and confirmed multivessel CAD with ≥70% stenosis, each IQR increase in IL-6, hsTnT, GDF-15, NT-proBNP, or sCD40L was individually associated with the primary and secondary outcome [63]
	IL-6 levels were higher in CAD patients than in the control group. Moreover, IL-6 levels were positively related to Gensini scores [36]
	Serum levels of IL-6 showed higher levels in the CAD group (8.67 ± 3.66 pg/mL) compared to the control group (5.39 ± 1.82 pg/mL; *p* < 0.001) [64]
	Serum TNF-α, and IL-6 levels significantly elevated in CAD group compared to the control group [65]
	The blood levels of IL-6 did not differ between patients with atherosclerosis and healthy controls [32]
	Serum IL-6 was found to be higher in the CAD group than in the control group [66]
	The concentrations of serum IL-25, IL-6 and TNF-α positively correlated with the Gensini Score [67]
	IP10, IL-6, and TNF-α levels in CAD patients were significantly higher than those in controls. Concerning positive relationship between miRNA 296-a gene expression level and serum concentrations of IL-6 and TNF-α in CAD patients, it is proposed that IL-6 and TNF-α inhibitor could be the main targets of miRNA 296a and, thereby, the IL-6 and TNF-α levels were increased; however, further study is needed [68]
	Serum concentration of IL-6 and TNF-a are significantly increased in the patients with CAD than the healthy controls [69]
	Unstable angina pectoris patients had lower serum IGF-1, IGFBP-4, and STC2 and higher serum inflammatory cytokine (hs-CRP, TNF-α, and IL-6) levels than the healthy controls [70]
	IL-6 levels were associated with the severity of CAD assessed by the GS. Moreover, based on the highest levels of IL-6 measured in patients with STEMI, the study strongly suggests that IL-6 could be a powerful marker in evaluating myocardial necrosis [71]
	In intermediate risk patients referred for coronary angiography, a serum IL-6 level above 1 pg/mL is predictive of significant CAD. IL-6 determination may be useful to reclassify CAD intermediate risk patients into higher risk categories [72]
	Plasma IL-6 levels are significantly associated with increased total and noncalcified short-term plaque progression in patients with stable coronary artery disease [34]
	None of the 10 cytokine levels, namely: IL-1β, IL-2, IL-4, IL-5, IL-6, IL-8, IL-10, TNF-α, granulocyte-macrophage colony stimulating factor (GM-CSF) and interferon-γ (IFN-γ), were correlated with the severity of CAD [9].
	IL-6 serum level did not change significantly between severe and non-severe CAD patients. IL-12p70 and IL-17, HDL-C, gender and diabetes were the independent predictors of severe CAD [10]

**Abbreviations:** CAD—coronary artery disease; CRP—C-reactive protein; CCTA—computed tomography of coronary arteries; CTO—chronic total occlusion; CVD—cardiovascular disease; ET-1—endothelin-1; GDF-15—growth differentiation factor-15; GS—Gensini Score; HDL-C—high-density lipoprotein cholesterol; hsCRP—high-sensitivity C-reactive protein; DM—Diabetes Mellitus; hs-cTn—high-sensitivity cardiac troponin; hsTnT—high-sensitivity troponin T; IGF-1—insulin-like growth factor 1; IGFBP-4—insulin-like growth factor binding protein 4; IL-1β—interleukin-1 beta; IL-2—interleukin-2; IL-4—interleukin-4; IL-5—interleukin-5; IL-6—interleukin-6; IL-8—interleukin-8; IL-10—interleukin-10; IL-12p70—interleukin-12p70; IL-17—interleukin-17; IL-25—interleukin-25; IL-32—interleukin-32; IL-36—interleukin 36; IQR—interquartile range; 8-iso-PGF2α—8-iso Prostaglandin F 2α; LVEF—left ventricular ejection fraction; MACEs—major adverse cardiovascular events; MetS—metabolic syndrome; NT-proBNP—N-terminal pro B-type natriuretic peptide; PAI-1—plasminogen activator inhibitor-1; PCAD—premature coronary artery disease; PCI—percutaneous coronary intervention; sCD40L—soluble CD40 ligand; sICAM-1—soluble intercellular adhesion molecule-1; sVCAM-1—soluble vascular cell adhesion molecule-1; STC2—stanniocalcin-2; STEMI—ST-elevation myocardial infarction; TNF-α—tumor necrosis factor-α.

## Data Availability

The raw data supporting the conclusions of this article will be made available by the authors on request.

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
