# Peer review of "Predictive Value of Selected Plasma Biomarkers in the Assessment of the Occurrence and Severity of Coronary Artery Disease"

_ijms, 2025, doi:10.3390/ijms26020537_

Round 1

Reviewer 1 Report

Comments and Suggestions for Authors

Dear Authors,

Thank you for raising the vital topic of biomarkers that could be predictive for CAD. Unfortunately, multiple major and minor methodological issues need to be addressed before a decision on the acceptance of your manuscript is made. 

1) The inclusion criteria are a little bit confusing, as you mentioned that in control groups, participants did not have stenosis greater than 50%. Does it mean that both patients had no stenosis (no cardiac atherosclerosis) and i.ex. 40% stenosis of coronary arteries? Also, you cannot call the control group non-CAD if participants in this group have CAD but with just a lower stenosis level.

2) Why did you set that stenosis level on 50%? Did you base it on the literature? If so, please cite the relevant papers. 

3) Why did you enroll 40 participants? Did you perform the power and sample size calculations? If so please provide the results, if not it has to be mentioned in study limitations.

4) Why did you pick such markers? Did you perform a wider screening or randomly choose those? You mentioned the literature review, but you omitted a large number of other groups of biomarkers that have been proven to be relevant in the aspect of CAD and also outcomes of coronary interventions (i.ex. leukotrienes - doi: 10.5603/AA.2022.0013)

5) Were the participants only examined by angiography, or did they also undergo angioplasty with stenting in the case of significant stenosis? 

6) How long after angiography were the blood samples taken?

Kind regards

Author Response

Thank you for your valuable comments, here are our answers:

1 The inclusion criteria are a little bit confusing, as you mentioned that in control groups, participants did not have stenosis greater than 50%. Does it mean that both patients had no stenosis (no cardiac atherosclerosis) and i.ex. 40% stenosis of coronary arteries? Also, you cannot call the control group non-CAD if participants in this group have CAD but with just a lower stenosis level.

The non-CAD group included patients without atherosclerosis or with mild intimal thickening. We have corrected the characterization of the non-CAD group in the text of the manuscript – lines 110-120 and 258-261.

The CAD group included patients with the so-called obstructive atherosclerosis.

2) Why did you set that stenosis level on 50%? Did you base it on the literature? If so, please cite the relevant papers.

50% stenosis is usually the cut-off point for hemodynamically significant atherosclerosis

https://academic.oup.com/eurheartjsupp/article/23/Supplement_C/C164/6357810

3) Why did you enroll 40 participants? Did you perform the power and sample size calculations? If so please provide the results, if not it has to be mentioned in study limitations.

For ADAMTS1, this is a pilot study, so test power and sample size could not be calculated.

The most important assays for us related to ET-1 and sVCAM-1, so we made the test power analysis and sample size for these two particles. Test power analysis and sample size (statistica 13.3) with the given concetration values revealed that a sample size of 5 or 10 individuals (VCAM-1 and ET-1 respectively) is sufficient with a test power of 0.9:

- For VCAM-1 (5 individuals in a group would be sufficient to demonstrate significance):

Sensitivity, specificity of biochemical markers for early prediction of endothelial dysfunction in atherosclerotic obese subjects - PMC

- For ET-1 (10 individuals in a group would be sufficient to demonstrate significance):

https://doi.org/10.5114/aoms.2019.86770

4) Why did you pick such markers? Did you perform a wider screening or randomly choose those? You mentioned the literature review, but you omitted a large number of other groups of biomarkers that have been proven to be relevant in the aspect of CAD and also outcomes of coronary interventions (i.ex. leukotrienes - doi: 10.5603/AA.2022.0013)

The aim of our study was to determine markers involved in the process of plaque formation at various stages of atherosclerosis development. We chose the relatively little studied sVCAM-1 and ET-1. Moreover we chose ADAMTS1, which to our knowledge no one has studied so far in the context of atherosclerosis. After analyzing the literature, we selected additional two promising cytokines: IL-6 and IL-8 – the information added in the text lines 140-144.

5) Were the participants only examined by angiography, or did they also undergo angioplasty with stenting in the case of significant stenosis?

The patient underwent only angiography – the information added in the manuscript lines 252-253

6) How long after angiography were the blood samples taken?

The blood was taken 2 hour after angiography - the infromation added in the manuscript line 263

Reviewer 2 Report

Comments and Suggestions for Authors

Thank you for submitting your manuscript titled "Predictive value of selected plasma biomarkers in the assessment of the occurrence and severity of coronary artery disease" for consideration. While the topic is of significance in the field of cardiovascular research, the manuscript in its current form has several critical shortcomings that need to be addressed to meet the publication standards. Below are detailed comments and suggestions for improvement:

1. The study design is inadequately described. Critical details such as the sampling method, inclusion and exclusion criteria, and ethical considerations are insufficient.

2. A total of 40 patients is a small sample size for this type of study. There is no discussion on sample size calculation or power analysis to justify whether this number is statistically adequate to detect significant differences.

3. There is a lack of control for potential confounding variables (e.g., age, gender, comorbidities, medications) that could influence biomarker levels.

4. The methods for biomarker assessment need more detail. Information on assay sensitivity, specificity, and validation should be provided.

5. The results section lacks clarity and depth. The data presentation is minimal, and essential details are missing.

6. There is no comprehensive description of the statistical methods used, such as tests for normality, handling of missing data, or correction for multiple comparisons.

7. The figures provided are not sufficiently informative. Figure 1 lacks detailed lengends and explanations necessary for interpretation.

8. The discussion does not critically interpret the findings in the context of existing literature. There is an overemphasis on IL-8 without adequately addressing why other biomarkers did not show significance.

9. The conclusion makes assertions that are not fully supported by the data, given the study's limitations.

I encourage you to consider these comments and substantially revise the manuscript to address the highlighted issues. A more robust study design, clearer presentation of methods and results, and a critical discussion will enhance the quality of your work and its suitability for publication.

Author Response

Thank you for your valuable comments, here are our answers.

1. The study design is inadequately described. Critical details such as the sampling method, inclusion and exclusion criteria, and ethical considerations are insufficient.

 More information added – lines 240-261

  1. A total of 40 patients is a small sample size for this type of study. There is no discussion on sample size calculation or power analysis to justify whether this number is statistically adequate to detect significant differences.

For ADAMTS1, this is a pilot study, so test power and sample size could not be calculated.

The most important assays for us related to ET-1 and sVCAM-1, so we made the test power analysis and sample size for these two particles. Test power analysis and sample size (statistica 13.3) with the given concetration values revealed that a sample size of 5 or 10 individuals (VCAM-1 and ET-1 respectively) is sufficient with a test power of 0.9:

- For VCAM-1 (5 people in a group would be sufficient to demonstrate significance):

Sensitivity, specificity of biochemical markers for early prediction of endothelial dysfunction in atherosclerotic obese subjects - PMC

- For ET-1 (10 individuals in a group would be sufficient to demonstrate significance):

https://doi.org/10.5114/aoms.2019.86770

  1. There is a lack of control for potential confounding variables (e.g., age, gender, comorbidities, medications) that could influence biomarker levels.

 Table 1 shows the patients characteristics. It was enriched in statin therapy and AF. There were no differences between CAD and non-CAD groups in most of the variable assessed except for smoking.

  1. The methods for biomarker assessment need more detail. Information on assay sensitivity, specificity, and validation should be provided.

ELISA assays used in this work were validated and their manufacturers, who also assessed their sensitivity, specificity. This information is available on the manufacturer's website. In our laboratory, we do not perform additional validations of purchased kits. The result of each assay was assessed on the basis of a standard curve drawn on the basis of protein standard concentrations that were validated by the manufacturer. Additional information on conducting the ELISA test was provided in lines 273-280 of the manuscript and in the table below:

Characteristics of ELISA tests used in the experiment with particular emphasis on their sensitivity, specificity and validation.

ELISA assay

sensitivity

specificity

validation

Endothelin-1 ELISA Kit

Thirty-four assays were evaluated and the minimum detectable dose (MDD) of Endothelin-1 ranged from 0.031-0.207 pg/mL. The mean MDD was 0.087 pg/mL. The MDD was determined by adding two standard deviations to the mean O.D. value of twenty zero standard replicates and calculating the corresponding concentration.

This assay recognizes natural and synthetic Endothelin-1. Human Endothelin-2 does not interfere but does cross-react approximately 23.4% in this assay.

Human/Rat Endothelin-3 does not interfere but does cross-react approximately 0.5% in this

assay.

Three samples of known concentration were tested twenty times on one plate to assess intra-assay precision.

Three samples of known concentration were tested in forty separate assays to assess

inter-assay precision. Assays were performed by at least three technicians using two lots of

components.

Human IL-6 ELISA Kit

The biological sensitivity of the assay is l2.813pg/ml. The range is 4.688-300pg/ml

Specifically binds with IL-6 , no obvious cross reaction with other analogues.

For inter-assay precision

samples with low, medium and high concentration are tested 20 times on the same plate.

For intra-assay precision

samples with low, medium and high concentration are tested 20 times on three different plates.

Human IL-8 ELISA Kit

The analytical sensitivity of the assay is <5.0 pg/mL human IL-8. This

was determined by adding two standard deviations to the mean O.D.

obtained when the zero standard was assayed 20 times.

Buffered solutions of a panel of substances at 50 ng/mL were assayed

with the Human IL-8 ELISA Kit. The following substances were tested

and found to have no cross-reactivity: human IL-1a, IL-1b, IL-1ra,

IL-2, IL-3, IL-4, IL-6, IL-7, IL-10, IFN-a, IFN-g, GM-CSF, OSM, MIP-1a,

MIP-1b, LIF, MCP-1, G-CSF, PF-4, bTG, GRO, IP-10, TNF-a, TNF-b,

TGF-b, and SCF. This IL-8 assay is able to recognize the 72 aa and 78 aa

forms of IL-8.

For inter-assay precision

samples were assayed 10 times in 5 different assays to determine

precision between assays.

For intra-assay precision

samples of known human IL-8 concentration were assayed in

replicates of 16 to determine precision within an assay.

  1. The results section lacks clarity and depth. The data presentation is minimal, and essential details are missing.

 More information added - Lines 110-130

  1. There is no comprehensive description of the statistical methods used, such as tests for normality, handling of missing data, or correction for multiple comparisons.

 The information added – lines 281-294

  1. The figures provided are not sufficiently informative. Figure 1 lacks detailed lengends and explanations necessary for interpretation.

The legend for fig. 1 was changed and fig 2 was added

  1. The discussion does not critically interpret the findings in the context of existing literature. There is an overemphasis on IL-8 without adequately addressing why other biomarkers did not show significance.

The discussion was changed – lines 178-184 added

  1. The conclusion makes assertions that are not fully supported by the data, given the study's limitations.

The conclusions are drawn on the basis of our own research and available literature and are deeply considered and, in our opinion, valuable so we would rather not change them.

Round 2

Reviewer 1 Report

Comments and Suggestions for Authors

Dear Authors,

Thank you for the responses and clarifications. Unfortunately, I noticed in the anti-plagiarism report that a huge part of the introduction is just a simple copy-paste of another article (https://academic.oup.com/cardiovascres/article/48/1/158/361640). This is unacceptable and must be rewritten, as in the present, for it is just a sample of scientific plagiarism.

Kind regards

Author Response

Thank you for the review and the important information about the anti-plagiarism report.
I wrote the introduction myself based on the literature.
If there are any sentences or fragments that look similar to other published ones,
I will of course change them, I just need an anti-plagiarism report so that I know
which fragments they are. F
or now, I've made some corrections to the introduction - yellow.
Thank you.

Reviewer 2 Report

Comments and Suggestions for Authors

Thank you for submitting the revised version of your manuscript. I appreciate your efforts to address my comments. 

However, after careful evaluation of the revised manuscript and your responses, we find that the manuscript still lacks the necessary scientific rigor in study design, methodology, and data interpretation. The efforts to address the my comments are minimal, and critical issues remain unresolved.

1. While you mention that this is a pilot study for ADAMTS1 and provide some references suggesting that small sample sizes may suffice for detecting significant differences in ET-1 and sVCAM-1 levels, this does not adequately justify the small sample size in your study. The power analysis provided is based on previous studies, not on your own data. A pilot study should still include a sample size calculation to ensure that the study is adequately powered to detect meaningful differences, especially when the conclusions drawn have clinical significance.

2. Despite the addition of Table 1 and mention that most variables did not differ between groups, smoking was significantly more common in the non-CAD group. Smoking is a well-known risk factor for CAD and could significantly influence biomarker levels. The lack of adjustment for this and potentially other confounders (e.g., statin therapy, atrial fibrillation) remains a critical flaw. Multivariate analysis should be conducted to adjust for these variables.

3. Collecting blood samples two hours after coronary angiography poses significant concerns due to the potential influence of the procedure on biomarker levels. You acknowledge in your response that this could have influenced the levels of the tested substances but do not provide any mitigation strategies or discuss this limitation adequately in the manuscript.

4. While you have added some information in lines 110-130, the results section still lacks depth. There is minimal statistical analysis presented, and the data are not thoroughly explored. For instance, you mention a positive correlation between IL-8 levels and the Gensini Score but do not provide confidence intervals or discuss the strength and implications of this correlation in detail.

5. The addition of lines 178-184 is noted; however, the discussion still lacks a critical analysis of why most biomarkers did not show significant differences. Potential reasons (e.g., sample size, patient selection, impact of timing of sample collection) should be explored in depth. Overemphasis on IL-8 without adequately addressing the negative findings for other biomarkers creates an imbalance in the discussion.

6. Your refusal to adjust the conclusions despite the identified limitations is problematic. Conclusions must be directly supported by the data presented and should be tempered to reflect the study's limitations. As it stands, the conclusions overstate the clinical utility of IL-8 as a diagnostic tool.

Author Response

Thank you very much for your insightful review. We have made major changes to the text and we believe that thanks to these comments, the text of the manuscript has gained a lot and is more valuable.

  1. While you mention that this is a pilot study for ADAMTS1 and provide some references suggesting that small sample sizes may suffice for detecting significant differences in ET-1 and sVCAM-1 levels, this does not adequately justify the small sample size in your study. The power analysis provided is based on previous studies, not on your own data. A pilot study should still include a sample size calculation to ensure that the study is adequately powered to detect meaningful differences, especially when the conclusions drawn have clinical significance.

We added the relevant information in methodology section lines 312-335

  1. Despite the addition of Table 1 and mention that most variables did not differ between groups, smoking was significantly more common in the non-CAD group. Smoking is a well-known risk factor for CAD and could significantly influence biomarker levels. The lack of adjustment for this and potentially other confounders (e.g., statin therapy, atrial fibrillation) remains a critical flaw. Multivariate analysis should be conducted to adjust for these variables.

Univariate analysis was performed and we added short information about it in the results section. However there are too little data for multivariate analysis – the information was added to the ‘limitations if the study’.

  1. Collecting blood samples two hours after coronary angiography poses significant concerns due to the potential influence of the procedure on biomarker levels. You acknowledge in your response that this could have influenced the levels of the tested substances but do not provide any mitigation strategies or discuss this limitation adequately in the manuscript.

Yes, we agree, this could have influenced the levels of the tested substances, but the study was a part of the larger project. We added the information in the ‘limitations of the study’.

  1. While you have added some information in lines 110-130, the results section still lacks depth. There is minimal statistical analysis presented, and the data are not thoroughly explored. For instance, you mention a positive correlation between IL-8 levels and the Gensini Score but do not provide confidence intervals or discuss the strength and implications of this correlation in detail.

We changed the results section, added some information in the table 1 and we changed the fig.2 description

  1. The addition of lines 178-184 is noted; however, the discussion still lacks a critical analysis of why most biomarkers did not show significant differences. Potential reasons (e.g., sample size, patient selection, impact of timing of sample collection) should be explored in depth. Overemphasis on IL-8 without adequately addressing the negative findings for other biomarkers creates an imbalance in the discussion.

We adapted to the suggestions and radically changed the discussions, focusing more on the analysis of the obtained results.

  1. Your refusal to adjust the conclusions despite the identified limitations is problematic. Conclusions must be directly supported by the data presented and should be tempered to reflect the study's limitations. As it stands, the conclusions overstate the clinical utility of IL-8 as a diagnostic tool.

We changed the conclusions.

Round 3

Reviewer 1 Report

Comments and Suggestions for Authors

Dear Authors,

As I received from the Editor the final version of the manuscript with a changed introduction, I have no further requests.

Kind regards

Reviewer 2 Report

Comments and Suggestions for Authors

The authors have not substantively addressed my concerns. However, the shortcomings of the paper have been acknowledged and discussed. If the paper is published, I still believe it will have some value to the academic community.